# Physical and Physiological Responses of U-14, U-16, and U-18 Soccer Players on Different Small-Sided Games

**DOI:** 10.3390/sports8050066

**Published:** 2020-05-18

**Authors:** Jorge López-Fernández, Javier Sánchez-Sánchez, Jorge García-Unanue, Enrique Hernando, Leonor Gallardo

**Affiliations:** 1Centre for Sport, Exercise and Life Science, Coventry University, Coventry CVI 5FB, UK; 2School of Sport Science, Universidad Europea de Madrid, 28670 Villaviciosa de Odón, Spain; javier.sanchez2@universidadeuropea.es; 3IGOID Research Group, University of Castilla-La Mancha, 45071 Toledo, Spain; jorge.garciaunanue@uclm.es (J.G.-U.); Enrique.Hernando@uclm.es (E.H.); Leonor.Gallardo@uclm.es (L.G.)

**Keywords:** activity profile, football, teenagers, youth

## Abstract

As most existing studies in youth academies are focused on top players, the objective of this research is to analyze the physical and physiological demands of various small-sided games (SSGs) on different age categories within a sub-elite soccer academy. We evaluated 63 young players from a Spanish sub-elite academy (under 14 = 21; under 16 = 21; under 18 = 21). Players performed four different small-side games focused on possession game (3-a-side; 4-a-side; 5-a-side; 6-a-side). The global indicators of performance and high-intensity actions were recorded through global positioning systems, whereas the heart rate responses were measured using heart rate monitors. Results: Under 16 ran a greater distance at high-intensity velocity than under 14 in the small side games 3v3 and 6v6. Furthermore, under 16 also ran a greater distance at high-intensity velocity than under 18 in the small side game 3v3 (*p* < 0.01). Under 14 showed greater acceleration at the highest intensity (> 2.75 m/s2) than the other age groups, under 16 and U18 (*p* < 0.01; ES (effect size) > 1). According to the physiological load, SSG 3v3 presented lower outcomes in Zone 6 (> 95% HRmax) than the small side game 4v4 and the small side game 5v5, in both under 14 and under 16. The workload of SSGs varies depending on the number of players, but also depending on the players’ ages. Therefore, when designing the SSGs it is important to consider both the players’ ages and the workload that want to be achieved.

## 1. Introduction

Soccer is one of the most participated-in sports in the world, and a very complex one, where players need technical, tactical and physical skills to achieve success [1]. Besides the technical and tactical components, aerobic fitness plays a key role in the players’ performance, as one of the main components for dealing with the physical demands of soccer games [2]. For this reason, how the body responds to the physical demands of the game—i.e., sprinting, running at different intensities with and without the ball, acceleration and deceleration (external and internal load)—are of great interest for the literature [3,4]. In this regard, the Global Positioning Systems (GPS) technology has become fundamental to quantify these kinds of loads. Due to its non-invasive nature, GPS can be used during games and training sessions, providing reliable measures that can be used to manage the load of the players [5,6]. Thus, this technology allows soccer staff to more accurately replicate in training the demands of the matches, and therefore, to support the player in developing the desired adaptations [7,8].

Within the youth academies, the small-sided games (SSGs) are widely practiced because they somehow replicate the physical, tactical and technical strain endured by players during a soccer match [9,10], but also because they allow the players to have a high number of contacts with the ball [11,12]. In this regard, previous studies have shown that the design of the SSG can enhance or reduce the number of contacts with the ball, the physiological demands, or the kinematic performance of soccer players [12,13]. Thus, for optimizing the training efficiency in youth academies, it is important to consider variables like the pitch size [14], the number of touches [15], the presence or absence of goalkeepers or small goal areas [16], the skill level of players [17] and the players’ ages [14,18]. Unfortunately, almost all the existing evidence comes from studies focused on adults or top older teenagers [9,19,20], so designing SSGs for non-elite young players according to the conclusions from these works may be inappropriate. Furthermore, the few works comparing different age categories evidence that each age-category may require different SSGs design [14,18]. 

The pitch size of SSGs is the most studied variable in youth, but surprisingly most of the studies only assess one age category, or use a different methodology that does not allow for making comparisons [13,14,21]. Moreover, although Olthof et al. [20] did assess the pitch size effect from U13 to U19, they did not compare among age groups. Therefore, the interaction-effect of each age category in SSGs remains unknown. This fact also occurs with other variables less studied, like the effect of the number of players, a variable which, despite being a key factor when designing SSGs in youth academies [13], just a few studies have assessed in young soccer players [19,22]. Moreover, it has only been examined in elite players, and never according to age category. Therefore, to understand how to design effective SSGs in youth academies, further research is required that assess the influence of the number of players in SSGs at the different developmental stages. For that reason, the main objective of the present study was to analyze the physical and physiological demands of various small-sided games (3v3, 4v4, 5v5, 6v6), and the interaction with different age categories (U14, U16, and U18) within a sub-elite soccer academy. 

## 2. Methods

### 2.1. Participants

We recruited 63 young Spanish soccer players from the same sub-elite academy for the study, and gathered them into three age groups: U14 (n = 21, height: 162.7 ± 10.0 cm, body mass: 50.9 ± 7.5 kg), U16 (n = 21, height: 168.5 ± 8.4 cm, body mass: 62.2 ± 9.0 kg), and U18 (n = 21, height: 172.0 ± 4.9 cm, body mass: 67.6 ± 8.2 kg). A total of 504 measurements were registered (two measures per player and SSG), but 24 were deleted due to errors in the recording process (these measures were partially recorded by the GPS devices). Thus 480 measures were included for the statistical analysis. Both players and their parents or legal guardians provided written informed consent. Participants and their coaches were informed in detail about this investigation and notified that they could withdraw from the study at any time. This research was approved by the local Clinical Research Ethical Committee in accordance with the Declaration of Helsinki.

### 2.2. Experimental Approach to the Problem

The study was conducted between April and May, within the competitive period. Each age category participated in the study during two consecutive weeks (4 days in total, 2 days each week). The temperature and relative humidity of all sessions oscillated between 24 and 28 °C and 38% and 49%, respectively. All tests were held during training time to avoid the influence of circadian cycles. Players agreed to rest 48 h prior to each test day (participants did not report intense physical activity or exercise in their physical education class at school during the test day). In the first day, participants completed a Yo-Yo Intermittent Endurance Test Level 2 to identify the maximum heart rate of each participant [23]. The second day was devoted to players completing a familiarization session to become accustomed to SSGs and the equipment used on the study. Finally, the main tests were conducted on the last two days during the second week (third and fourth days).

The main test consisted of performing four different SSGs (3v3, 4v4, 5v5, 6v6) at four minute intervals. Each SSG had different pitch dimensions but similar area per player (Table 1). A four minute passive recovery was established between SSGs and players were encouraged to drink water within the recovery time. On the third day players played the SSGs in the following order: 3v3, 4v4, 5v5 and 6v6. On the fourth day, players repeated the same SSGs, but the order was reversed (6v6, 5v5, 4v4, 3v3) to reduce the influence of fatigue or motivation and increase data reliability.

All sessions began with the same standardized warm-up consisting of five minutes of continuous running, five minutes of joint mobility exercises and three sprints at increasing speed [24]. The participants’ physical responses (global indicators of performance, sprints performance, and acceleration and deceleration profiles) were collected though GPS (HPU; GPSports, Canberra, Australia), whereas the physiological responses were monitored with heart rate monitors (Polar Electro, Kempele, Finland). Following the manufacturer’s recommendations, all GPS devices were activated at least 15 min before the start of the investigation. Only data information provided by at least eight satellites or more were included in the study. Data collection and processing took place through the manufacturer’s specialized software (Team AMS, version 2016; GPSports, Canberra, Australia).

### 2.3. Procedures

#### 2.3.1. Global Indicators of Performance

The GPS devices provided the following data: total distance of each SSG (TD); meters covered per minute (Mean m/min); peak speed (V_max_ Peak); and average speed (V_mean_). The players’ physical profiles were established through six speed ranges [25]: walking/walking fast (from 0 to 6.9 km/h); jogging (7.0 to 9.9 km/h); running (10.0 to 12.9 km/h); fast running (from 13.0 to 15.9 km/h); high-speed running (16.0 to 17.9 km/h); and sprint (> 18.0 km/h).

#### 2.3.2. Sprint Performance Variables

High-speed actions (HI) (all actions above 16 km/h) were analyzed in detail. The following values were recorded: high-intensity distance (m) and high-intensity as percentage of total distance (%DT); number of high-intensity actions (n); average duration of sprints (s); average maximum speed (km/h); average distance of sprint (m); and acceleration max mean (Acc max; m/s^2^).

#### 2.3.3. Acceleration and Deceleration Profile

Four zones were established for both accelerations and decelerations: 1.5–2.0 m/s^2^; 2.0–2.5 m/s^2^; 2.5–2.75 m/s^2^; and > 2.75 m/s^2^ [1].

#### 2.3.4. Physiological Responses

Based on the maximum heart rate (HR_max_) obtained in the Yo-Yo Intermittent Endurance Test Level 2, six zones of intensity were defined as in the previous research [26]: Zone 1 (< 60% HR_max_); Zone 2 (60%–70% HR_max_); Zone 3 (70%–80% HR_max_); Zone 4 (80%–90% HR_max_); Zone 5 (90%–95% HR_max_); and Zone 6 (> 95% HR_max_). Data are presented as relative percentages in relation to the total played time. Furthermore, the heart rate peak (HR_peak_) and average heart rate (HR_mean_) in beats per minute (b.p.m.), and the percentage of maximum individual heart rate (%HR_max_), were also recorded.

### 2.4. Statistical Analysis

Data are presented as mean ± standard deviation. The coefficient of variation (%) was obtained through all significant variables according to age groups. A two-way ANOVA and Bonferroni post-hoc were performed to identify the difference in the total distance travelled, patterns of movement, heart rate, and work and rest ratios, among different age groups and SSGs. Confidence interval (CI of 95%) was included to identify the magnitude of changes. Effect sizes (ES) were calculated and defined as follows: null, < 0.3; mild, 0.3–0.5; moderate, 0.5–0.7; strong, 0.7–0.9; and very strong, 0.9–1.0 [27]. All calculations were carried out using SPSS software (version 20.0, Chicago, IL, USA). Statistical significance was set at *p* < 0.05.

## 3. Results

### 3.1. Global Indicators of Performance

The outcomes of global indicators of performance are displayed in Table 2. In the SSG 3v3, the U14 covered less distance per minute (Mean m/min) and achieved lower average speed (V_mean_) than the older groups (*p* < 0.05). Moreover, they showed lower values in the SSG 3v3 than U16 (−46.5 m; *p* = 0.001; ES: 1.10; CI: −18.3 to −74.7) and U18 (−33.8 m; *p* = 0.01; ES: 0.86; CI: −5.56 to −62.0). In the SSG 6v6, U16 covered greater distance per minute and higher TD than U14 and U18 (*p* < 0.05). On the other hand, all age groups presented higher TD and distance per minute in the SSG 3v3 than the SSG 6v6 (*p* < 0.05). The older groups, U16 and U18, presented higher TD in the SSG 3v3 than 5v5 (*p <* 0.05).

Figure 1 presents the players’ physical profile. For the SSG 3v3, U16 players ran more distance in Zone 5 (16.0 to 17.9 km/h) than U14 (+26.29 m; *p* = 0.001; ES: 1.18; CI: 14.71 to 37.87) and U18 (+12.32 m; *p* = 0.03; ES: 0.65; CI: 0.74 to 23.90). Regarding Zone 6, the distance covered in the SSG 5v5 was higher than the distance covered in the SSG 4v4, both for U16 (+5.56 m; *p* = 0.01; ES: 0.35; CI: 0.75 to 10.37) and U18 (+5.06 m; *p* = 0.03; ES: 0.36; CI: 0.25 to 9.87).

### 3.2. Sprint Performance Variables

Table 3 shows the players’ performance in sprint actions. The U16 ran a greater distance at high intensity than U14 (*p* < 0.01) in the SSG 3v3 and the SSG 6v6, and greater distance at high intensity than U18 in the SSG 3v3 (+23.1 m; *p* = 0.003; ES: 0.78; CI: 6.4 to 39.8). Moreover, in the SSG 6v6 the U16 also displayed a higher number of sprints than the U14 (+0.43 m; *p* = 0.04; ES: 0.46; CI: 0.005 to 0.85) and the U18 (+0.43; *p* = 0.03; ES: 0.42; CI: 0.02 to 0.85). Regarding the U14, they achieved lower Acc max mean in sprint than both the U16 (−0.72 m/s^2^; *p* = 0.001; ES: 0.68; CI: −0.33 to −1.12) and the U18 (−0.72 m/s^2^; *p* = 0.001; ES: 0.67; CI: −0.32 to −1.11). In addition, they showed lower Acc max mean in the SSG 6v6 than the other three SSG formats (*p* < 0.05). Finally, the U14 also ran greater distances at high intensity in the SSG 5v5 than in the SSG 3v3, and higher average distance in sprint in the SSG 5v5 than in the SSG 4v4 (*p* < 0.05).

### 3.3. Acceleration and Deceleration Profile

The results of the acceleration and deceleration profiles are displayed in Table 4. The U14 performed greater acceleration and deceleration at the highest intensity (> 2.75 m/s^2^) than both U16 and U18 (*p* < 0.01; ES > 1). On the contrary, the U16 presented a higher number of accelerations between 2.5 and 2.75 m/s^2^ than the U14, for the SSG 3v3, SSG 4v4, and SSG 5v5 (*p* < 0.05), as well as greater decelerations between 2.5 and 2.75 m/s^2^ for the SSG 3v3 (+2.58 n; *p* = 0.001; ES: 0.68; CI: 0.81 to 4.34) and the SSG 6v6 (+1.87 n; *p* = 0.03; ES: 0.63; CI: 0.10 to 3.63).

When comparing among SSGs formats, the U14 and U16 got a higher number of accelerations between 2.5 and 2.75 m/s^2^ in the 3v3 than in both the SSG 5v5 and the SSG 6v6 (*p* < 0.05). Moreover, both the U16 and the U18 showed a greater number of accelerations at the lowest intensity (1.5 and 2.0 m/s^2^) in the SSG 3v3 than in the SSG 4v4 (U16 (+27.71 m; *p* = *0.029;* ES: 0.86; CI: 0.172 to 5.145); U18 (+27.71 m; *p* = *0.027;* ES: 0.86; CI: 0.196 to 5.169)]. Regarding decelerations, the U14 also presented a higher number of decelerations (*p* < 0.05) between 2.0 and 2.75 m/s^2^ among the smaller SSGs (3v3 and 4v4) and the biggest ones (5v5 and 6v6). The U16 achieved a higher number of decelerations between 2.5 and 2.75 m/s^2^ in the SSG 3v3 than the SSG 4v4 (+27.71 m; *p* = *0.011;* ES: 0.86; CI: 0.357 to 4.179), while the U18 showed greater decelerations between 2.5 and 2.75 m/s^2^ in the SSG 4v4 than the SSG 6v6 (+27.71 m; *p* = *0.001;* ES: 0.86; CI: 0.772 to 4.594). Finally, both the U16 and the U18 showed a higher number of actions between 1.5 and 2.75 m/s^2^ in the SSG 3v3 than the SSG 6v6 (*p* < 0.05).

### 3.4. Physiological Responses

The players’ physiological responses are displayed in Table 5. When comparing the HR_mean_ and HR_peak_, either as b.p.m. or %HR_max_, U14 presented higher outcomes in the SSGs 4v4 and SSG 5v5 than in the SSG 3v3 (*p* < 0.05). The U18 showed higher HR_mean_ in the SSG 4v4 than in the SSG 5v5 (+3.37% HR_max_; *p* = 0.039; ES: 0.71; CI: 0.1 to 6.4) and higher HR_peak_ in the SSG 4v4 than in the SSG 6v6 (+1.17% HR_max_; *p* = 0.12; ES: 0.82; CI: 0.46 to 5.64). On the other hand, the U14 and U16 presented higher HR_mean_ and HR_peak_ in the SSGs 5v5 and SSG 6v6 than U18 (*p* < 0.05), while U16 displayed greater HR_peak_ in the SSG 3v3 than U14 (+2.8% HR_max_; *p* = 0.21; ES: 1.14; CI: 0.12 to 5.28).

Figure 2 shows the internal load of players as the percentage of time spent in each of the six intensity zones. When comparing the findings among ages, U16 displayed higher values than U18 in Zone 5 (SSG 3v3 (+21.02%; *p* = 0.003; ES: 1.07; CI: 5.80 to 36.24); SSG 5v5 (+14.97%; *p* = 0.042; ES: 0.72; CI: 0.37 to 29.58); SSG 6v6 (+17.33%; *p* = 0.02; ES: 0.84; CI: 2.054 to 32.58)) and Zone 6 (SSG 4v4 (+15%; *p* = 0.029; ES: 0.55; CI: 1.16 to 28.84); SSG 5v5 (+16.72%; *p* = 0.011; ES: 0.75; CI: 2.99 to 30.43); SSG 6v6 (+19.15%; *p* = 0.004; ES: 1.06; CI: 4.81 to 33.48)). Moreover, in Zone 5, the U18 also got lower values than U14 (SSG 5v5 (−17.65%; *p* = 0.01; ES: 0.90; CI: −32.09 to −3.22); SSG 6v6 (−17.46%; *p* = 0.015; ES: 0.77; CI: −32.34 to −2.57)). On the other hand, when comparing among SSGs, the SSG 3v3 presented lower outcomes in Zone 6 than the SSG 4v4 and the SSG 5v5 for both U14 (SSG 3v3 < SSG 4v4 (−15.43%; *p* = 0.049; ES: 1.51; CI: −30.83 to −0.27); 3v3 < 5v5 (−15.91%; *p* = 0.039; ES: 1.85; CI: −31.31 to −0.50)] and U16 (SSG 3v3 < SSG 4v4 (−19.22%; *p* = 0.009; ES: 0.91; CI: −35.15 to −3.28); SSG 3v3 < SSG 5v5 (−17.12%; *p* = 0.028; ES: 0.88; CI: −33.05 to −1.18)).

## 4. Discussion

To the best of the authors’ knowledge, this is the first work analyzing the physical and physiological demands of various small-sided games (SSGs) on different age categories within a sub-elite soccer academy. Previous studies have evidenced that the number of players can be used to manipulate the intensity of SSGs in youth [22,28,29]. However, these studies were focused on SSGs with goalkeepers [28] or goalposts [22], and only assessed a unique age category [22,28,29]. Moreover, most of them were focused on top youth players, whose findings somehow might differ from young, sub-elite soccer players [14,22,28,29].

When analyzing the physical responses of performance, we found that the 3v3 elicited greater total distance and m/min than 6v6 for all age groups, and 5v5 for U16 and U18; this was mainly due to young players running higher distance in Zone 2 (7−10 km/h) and Zone 3 (10−13 km/h) in the 3v3 than in the 5v5 and 6v6. The 4v4 also showed higher distance than the 5v5 and 6v6 in these zones, but with significant differences only for U18, probably because four minutes of play was too low to cause differences in the other younger groups. Some authors, like Hill-Haas et al. [22] and Brandes et al. [19], did not find differences in the TD when changing the number of players, concluding that TD is an inappropriate indicator of the work rate, as it is not sensitive to changes in number of players. However, they used a different approach, including goalposts or elite young players. Our outcomes suggest that this variable is sensitive to changes in number of players in possession games, and the different conclusions to those of Hill-Haas et al. [22] highlight the difficulty of controlling the intensity of SSGs, and the need for studying the influence of different variables rather than isolating one variable. Conversely, the higher speeds achieved by U16 and U18, over U14, are in line with the findings of Mendez-Villanueva et al. [30] and Nikolaidis et al. [31], who suggested that the speed ability of young players showed a greater increase at the beginning of adolescence than at the end. This can also explain the better performance of U16 than U14 in the high-intensity actions of 3v3 and 6v6.

Surprisingly, U16 displayed greater high-intensity performance during 3v3 (high-intensity distance) and 6v6 (number of sprints) than U18, despite the fact that both age groups presented similar run speeds. This is probably because U18 have a keener ability of “reading the game”, and exhibit higher cooperation among team players as a result of being more experienced [32]. Furthermore, the greater capacity of experienced players in maintaining ball possession in SSG, and keeping a wider dispersion on the pitch, may explain the lower TD values in U14 than in the remaining age groups, as well as the lower speed distance of U14, from 10 km/h onwards [20,32]. Finally, only the U14 showed differences in high-intensity actions among SSG formats, which may suggest that younger players are more sensitive to changes in the number of players [33]. The lack of significant differences among SSGs types for the sprint variables in the remaining age groups suggests that four minutes of play is too short for U16 and U18 to obtain different, high-intensity responses when modifying the number of players [9].

On the other hand, the number of accelerations increased in those SSGs with fewer players. This is probably due to the fact that participation in the game increases when the number of players decreases [28], and because young players seem to perform more technical actions in 3v3 than 6v6 [34]. The greater acceleration over 2.75 m in U14 than in the other two groups may be explained by the growth level. The U14 players were less physically developed than the other groups, so for the same number of players, the exercise load was higher in U14 than in the remaining groups [30,31]. This also explains the higher number of decelerations in U14 than in U16 and U18. Finally, the higher number of decelerations in those SSGs with fewer players suggests that these games have a higher eccentric component than those with a higher number of players [35,36]; therefore, coaches should avoid using these games for recovery.

When focusing on the physiological responses of young players to SSGs, several studies have considered that these games are useful tools for conditioning soccer players [13,21]. In this study, U14 and U16 presented higher HR_max_ and HR_mean_ than U18, indicating that for the same number of participants, the internal load was higher in younger players. Moreover, U14 and U16 spent more time in Zone 5 and Zone 6 than U18. This is probably due to the fact that younger players are less experienced, and therefore have a lesser ability to “read the game” and use the available space [18]. The growth peak in U16 players can also cause an increase in the internal load of SSGs [37]. Finally, the greater internal load in SSG 4v4 and SSG 5v5 than the SSG 3v3 in U14 suggests a greater physiological workload in those games than in 3v3, but only for U14. It may be caused by an increase in possession due to having more passing options in those game formats [13]. However, as this variable was not measured, further studies are required to verify this hypothesis. These outcomes do not match with those of Hill-Haas et al. [22], who found lower physiological workloads in young players when the number of players increased. However, they used U19 players and a different methodology (more playing time and different pitch size), suggesting that findings in U18 or U19 players should not be applied to younger players.

One limitation of this study is the duration of the SSGs. We selected four minute SSG formats because this duration has been proven sufficient to find differences in both physical responses [24], and studies have evidenced a reduction in heart rate intensity after six minutes of play in SSGs of 3v3 [38]. On the other hand, it is important to mention that this study uses measures from top youth players or adults to facilitate comparisons with other studies. However, we acknowledge that this fact somehow might influence the outcomes of this work. Moreover, most of the existing studies conducted in young players use a different methodology (e.g., goalpost or keepers vs possession games, or different pitch sizes, game duration and players’ level) and are focused on top players, so care must be taken when comparing our outcomes with previous studies [13,22]. Finally, it is important to highlight that it was not possible to randomize the order of the SSGs. However, to reduce the risk of bias, all age categories followed the same order. Moreover, in the first instance, the players completed the SSGs in the following order: 3v3, 4v4, 5v5 and 6v6; while the second time the players followed the inverse order (6v6, 5v5, 4v4 and 3v3).

The proper selection of the SSG for training in grassroots sport will allow better adaptation of the training to the demands of the match [7], thus reducing the risk of injury and improving the physical condition of the players, an essential element for their correct development and learning [39].

## 5. Conclusions

The internal and external loads demanded by the SSGs are different according to the age category. Therefore, coaches should consider this fact when designing SSGs for U14, U16 and U18. The intensity of SSGs can be manipulated by increasing or decreasing the number of players, U14 being more sensitive to changes in this variable than the other two groups.

## Figures and Tables

**Figure 1 sports-08-00066-f001:**
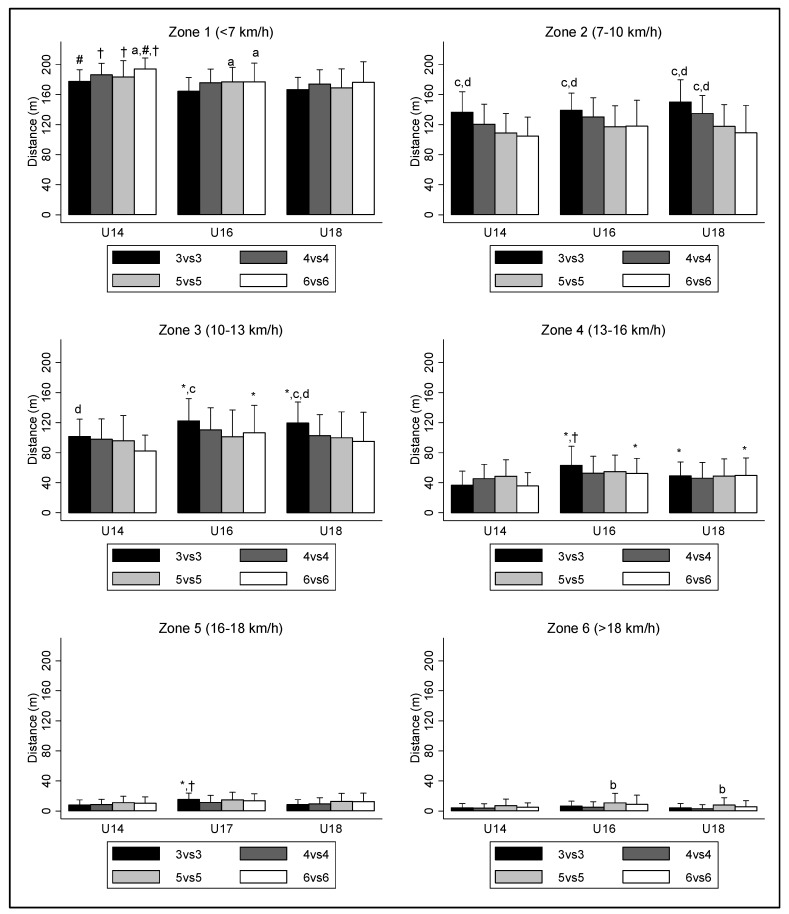
Distance zones for each SSG format and for each age category. a: Significant higher outcomes comparing to SSG 3v3 (*p* < 0.05); b: Significant higher outcomes comparing to SSG 4v4 (*p* < 0.05); c: Significant higher outcomes comparing to SSG 5v5 (*p* < 0.05); d: Significant higher outcomes comparing to SSG 6v6 (*p* < 0.05). *: Significant higher outcomes comparing to U14 (*p* < 0.05); #: Significant higher outcomes comparing to U16 (*p* < 0.05); †: Significant higher outcomes comparing to U18 (*p* < 0.05).

**Figure 2 sports-08-00066-f002:**
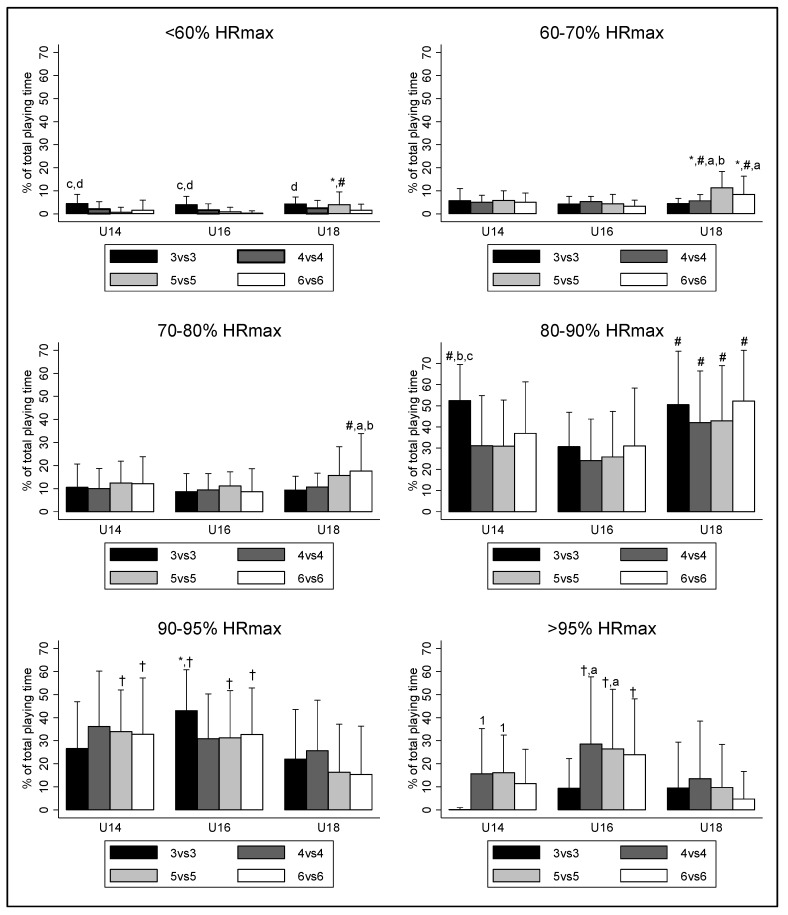
Physiological responses for each SSG format and for each age category. a: Significant higher outcomes comparing to SSG 3v3 (*p* < 0.05); b: Significant higher outcomes comparing to SSG 4v4 (*p* < 0.05); c: Significant higher outcomes comparing to SSG 5v5 (*p* < 0.05); d: Significant higher outcomes comparing to SSG 6v6 (*p* < 0.05). *: Significant higher outcomes comparing to U14 (*p* < 0.05); #: Significant higher outcomes comparing to U16 (*p* < 0.05); †: Significant higher outcomes comparing to U18 (*p* < 0.05).

**Table 1 sports-08-00066-t001:** Characteristics of small-sided games.

Design of SSG	Pitch Dimensions	Area Per Player (m^2^)	Duration	Recovery Time
3v3	30 × 20 m	100 m^2^	4 min	4 min
4v4	32 × 25 m	100 m^2^	4 min	4 min
5v5	37 × 27 m	99.9 m^2^	4 min	4 min
6v6	30 × 40 m	100 m^2^	4 min	4 min

**Table 2 sports-08-00066-t002:** Global indicator of performance among age groups and small-sided games (SSGs).

Variable	U14 (*)				U16 (‡)				U18 (†)			
	3v3 (a)	4v4 (b)	5v5 (c)	6v6 (d)	3v3 (a)	4v4 (b)	5v5 (c)	6v6 (d)	3v3 (a)	4v4 (b)	5v5 (c)	6v6 (d)
Total Distance (m)	464.2 (35.3) ^d^	462.3 (51.2)	454.8 (55.7)	432.1 (47.0)	510.7 (48.9) *^,c, d^	485.0 (42.0)	475.5 (52.9)	475.9 (49.8) *^,+^	497.9 (42.4) *^,c,d^	470.0 (43.8)	455.9 (63.2)	448.2 (78.31)
Mean m/min (m/min)	116.0 (8.8) ^d^	115.6 (12.8)	113.7 (13.9)	108.0 (11.8)	127.7 (12.23) *^,c,d^	121.3 (10.5)	118.9 (13.2)	118.9 (12.4) *	124.5 (10.6) *^,c,d^	117.5 (10.9)	113.9 (15.8)	112.1 (19.6)
V_peak_ (km/h)	18.16 (2.50)	18.31 (1.91)	19.03 (2.11)	18.70 (2.13)	19.32 (1.81) *	18.76 (1.99)	19.81 (2.63)	19.14 (2.14)	18.30 (1.99)	18.10 (1.65)	19.24 (2.41)	18.72 (2.16)
V_mean_ (km/h)	4.54 (0.62)	4.58 (0.47)	4.76 (0.52)	4.68 (0.53)	4.83 (0.45) *	4.69 (0.49)	4.95 (0.65)	4.78 (0.53)	4.58 (0.49) *	4.52 (0.41)	4.81 (0.60)	4.68 (0.54)

^a^: Significant higher outcomes (*p < 0.05)* comparing to SSG 3v3 (3-a-side); ^b^: Significant higher outcomes (*p < 0.05)* comparing to SSG 4v4(4-a-side); ^c^: Significant higher outcomes (*p < 0.05)* comparing to SSG 5v5 (5-a-side); ^d^: Significant higher outcomes (*p < 0.05)* comparing to SSG 6v6 (6-a-side). *: Significant higher outcomes (*p* < 0.05) comparing to U14; ^‡^: Significant higher outcomes (*p* < 0.05) comparing to U16; ^†^: Significant higher outcomes (*p* < 0.05) comparing to U18. Mean m/min: Mean of m covered per minute.

**Table 3 sports-08-00066-t003:** Sprint performance variables among age groups and small-sided games.

Variable	U14 (*)				U16 (‡)				U18 (†)			
	3v3 (a)	4v4 (b)	5v5 (c)	6v6 (d)	3v3 (a)	4v4 (b)	5v5 (c)	6v6 (d)	3v3 (a)	4v4 (b)	5v5 (c)	6v6 (d)
High Intensity Distance (%DT)	10.4 (5.3)	12.2 (5.1)	14.3 (5.8) ^a^	11.5 (5.4)	16.3 (5.6) *^,†^	14.0 (6.3)	16.6 (6.5)	15.6 (6.5) *	12.2 (4.63)	12.1 (5.5)	14.7 (6.88)	14.6 (6.7)
High intensity distance (m)	48.6 (25.3)	57.8 (27.2)	66.7 (31.5)	50.9 (26.4)	84.76 (33.8) *^,†^	68.9 (33.2)	80.3 (32.9)	74.7 (33.0) *	61.7 (25.45)	58.3 (29.7)	69.5 (36.7)	67.6 (37.1)
Number of Sprints (n)	0.32 (0.53)	0.21 (0.47)	0.53 (0.73)	0.45 (0.60)	0.59 (0.77)	0.44 (0.81)	0.85 (0.99)	0.88 (1.23) *^,†^	0.39 (0.63)	0.24 (0.58)	0.66 (0.88)	0.44 (0.84)
Sprint Vmax mean (km/h)	6.09 (9.72)	3.81 (8.13)	8.87 (10.70)	7.02 (9.92)	9.02 (10.36)	6.53 (9.72)	11.95 (10.75)	10.04 (10.44)	6.47 (9.63)	3.96 (8.15)	9.15 (10.50)	5.65 (9.47)
Sprint average distance (m)	2.8 (4.9)	1.6 (3.9)	4.5 (6.0)^b^	3.1 (4.7)	3.6 (4.5)	2.4 (3.7)	4.9 (5.1)	4.2 (4.8)	2.6 (4.0)	1.5 (3.2)	3.9 (4.9)	2.5 (4.3)
Acceleration max mean	2.74 (0.50) ^d^	2.41 (0.96) ^d^	2.69 (0.85) ^d^	1.93 (1.34)	2.85 (0.19)	2.77 (0.49)	2.85 (0.22)	2.65 (0.77) *	2.88 (0.19)	2.76 (0.66)	2.63 (0.90)	2.65 (0.78) *

^a^: Significant higher outcomes comparing to SSG 3v3 (*p* < 0.05); ^b^: Significant higher outcomes comparing to SSG 4v4 (*p* < 0.05); ^c^: Significant higher outcomes comparing to SSG 5v5 (*p* < 0.05); d: Significant higher outcomes comparing to SSG 6v6 (*p* < 0.05). *: Significant higher outcomes comparing to U14 (*p* < 0.05); ^‡^: Significant higher outcomes comparing to U16 (*p* < 0.05); ^†^: Significant higher outcomes comparing to U18 (*p* < 0.05).

**Table 4 sports-08-00066-t004:** Acceleration and deceleration profile of players among age group and SSG.

Variable	U14 (*)				U16 (‡)				U18 (†)			
	3v3 (a)	4v4 (b)	5v5 (c)	6v6 (d)	3v3 (a)	4v4 (b)	5v5 (c)	6v6 (d)	3v3 (a)	4v4 (b)	5v5 (c)	6v6 (d)
Accelerations												
Accl. 1.5 m/s^2^–2 m/s^2^ (n)	16.13 (5.50) ^c,d^	15.11 (4.66)	13.16 (4.21)	13.18 (3.82)	16.93 (4.89) ^b,c,d^	14.27 (4.27) ^d^	12.76 (3.87)	12.27 (3.64)	16.93 (4.41) ^b,c,d^	14.24 (4.02) ^d^	12.54 (3.43)	10.95 (3.95)
Accl. 2.0 m/s^2^–2.5 m/s^2^ (n)	9.74 (3.25) ^d^	9.42 (3.12) ^d^	8.13 (3.60)	7.01 (2.79)	11.17 (3.19) ^c,d^	10.76 (3.18) ^c,d^	8.24 (3.49)	8.34 (2.75)	11.59 (3.44) *^,c,d^	10.49 (3.33) ^c,d^	8.41 (3.29)	7.46 (3.38)
Accl. 2.5 m/s^2^–2.75 m/s^2^ (n)	5.21 (2.57) ^c,d^	4.50 (2.40)	3.82 (2.25)	3.63 (2.20)	7.15 (2.73) *^,c,d^	6.29 (2.93) *^,d^	5.10 (1.61) *	4.56 (1.95)	6.95 (2.45) *	5.46 (2.01)	4.22 (1.80)	3.93 (1.82)
Accl. >2.75 m/s^2^ (n)	1.50 (1.94) ^‡,†^	1.63 (1.82) ^‡,†d^	1.37 (1.65) ^‡,†^	0.95 (1.39)	0.20 (0.40)	0.20 (0.40)	0.20 (0.46)	0.12 (0.33)	0.20 (0.40)	0.34 (0.73)	0.24 (0.54)	0.20 (0.40)
Decelerations												
Dec. 1.5 m/s^2^–2 m/s^2^ (n)	13.16 (4.19)	11.45 (3.10)	11.37 (3.55)	11.18 (3.63)	14.56 (3.55) ^c,d^	12.90 (4.45) ^d^	11.10 (3.60)	10.71 (3.89)	13.83 (3.99) ^c,d^	13.39 (3.38) ^c,d^	11.05 (3.43)	11 (3.71)
Dec. 2.0 m/s^2^–2.5 m/s^2^ (n)	10.11 (3.73) ^+,d^	11.08 (3.51) ^c,d^	8.26 (3.58)	7.76 (3.34)	12.68 (3.72) *^,c,d^	11.12 (2.98)	9.12 (3.43)	9.12 (2.80)	12.54 (3.54) ^c,d^	10.93 (3.42) ^c,d^	8.73 (3.87)	7.90 (3.79)
Dec. 2.5 m/s^2^–2.75 m/s^2^ (n)	10.39 (3.76) ^c,d^	8.95 (3.51) ^d^	7.66 (3.53)	6.45 (3.08)	12.98 (3.83) *^,†,b,c,d^	10.71 (3.70) ^d^	8.95 (2.77)	8.32 (2.81) *	11.07 (3.24) ^c,d^	9.41 (2.93) ^d^	8.32 (3.07)	6.73 (2.75)
Dec. >2.75 m/s^2^ (n)	5.97 (6.06) ^‡,+,d^	4.89 (5.09) ^‡,+^	4.39 (4.55) ^‡,+^	3.45 (3.69) ^‡,+^	0.54 (0.74)	0.29 (0.56)	0.54 (0.67)	0.46 (0.92)	0.27 (0.50)	0.37 (0.73)	0.44 (0.71)	0.34 (0.53)

^a^: Significant higher outcomes comparing to SSG 3v3 (*p* < 0.05); ^b^: Significant higher outcomes comparing to SSG 4v4 (*p* < 0.05); ^c^: Significant higher outcomes comparing to SSG 5v5 (*p* < 0.05); ^d^: Significant higher outcomes comparing to SSG 6v6 (*p* < 0.05). *: Significant higher outcomes comparing to U14 (*p* < 0.05); ^‡^: Significant higher outcomes comparing to U16 (*p* < 0.05); ^†^: Significant higher outcomes comparing to U18 (*p* < 0.05). Accl: acceleration. Dec: deceleration.

**Table 5 sports-08-00066-t005:** Physiological responses of players among age groups and small-sided games.

Variable	U14 (*)				U16 (‡)				U18 (†)			
	3v3 (a)	4v4 (b)	5v5 (c)	6v6 (d)	3v3 (a)	4v4 (b)	5v5 (c)	6v6 (d)	3v3 (a)	4v4 (b)	5v5 (c)	6v6 (d)
HR mean (%HR_mean_)	83.44 (8.45)	87.16 (4.15) ^a^	87.12 (3.74) ^†,a^	86.21 (4.72) ^†^	86.61 (3.86)	88.18 (4.45)	88.31 (4.24) ^†^	88.91 (3.83) ^†^	84.72 (4.25)	85.87 (4.28) ^c^	82.50 (5.22)	82.74 (5.36)
HR mean (b.p.m.)	169.26 (8.45)	177.09 (8.48) ^a^	177.09 (9.47) ^†,a^	175.23 (11.29) ^†^	172.71 (9.77)	175.73 (11.08)	176.00 (10.92) ^†^	177.45 (9.58) ^†^	169.26 (7.55)	171.84 (7.70)	165.50 (9.36)	165.88 (9.13)
HR peak (%HR_max_)	92.09 (2.63)	95.10 (3.20) ^a^	95.99 (3.01) ^†,a^	94.49 (3.75) ^†^	94.89 (2.27) *	96.07 (3.11)	96.13 (3.54) ^†^	96.34 (3.19) ^†^	92.42 (3.88)	93.86 (3.65) ^d^	92.69 (4.13)	90.82 (3.80)
HR peak (b.p.m)	186.87 (7.94)	193.26 (7.71) ^†,a^	195.09 (8.05) ^†,a^	192.05 (9.84) ^†^	189.19 (6.93)	191.41 (8.16)	191.55 (9.48) ^†^	192.25 (8.22) ^†^	184.65 (6.57)	187.84 (5.89)	185.96 (6.50)	182.13 (6.40)

^a^: Significant higher outcomes comparing to SSG 3v3 (*p* < 0.05); ^b^: Significant higher outcomes comparing to SSG 4v4 (*p* < 0.05); ^c^: Significant higher outcomes comparing to SSG 5v5 (*p* < 0.05); ^d^: Significant higher outcomes comparing to SSG 6v6 (*p* < 0.05). *: Significant higher outcomes comparing to U14 (*p* < 0.05); ^‡^: Significant higher outcomes comparing to U16 (*p* < 0.05); ^†^: Significant higher outcomes comparing to U18 (*p* < 0.05). HR: Heart Rate. HRmax: Maximum Heart Rate.

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
