# Peer review of "Physical and Physiological Responses of U-14, U-16, and U-18 Soccer Players on Different Small-Sided Games"

_sports, 2020, doi:10.3390/sports8050066_

Round 1

Reviewer 1 Report

This study is well-thought and carried out. The authors should be commended for the standard and number of players that were assessed as well as the quality of the study.

There are several things I believe they should adjust to improve the manuscript:

Introduction:

- pg 2

  • The "why?" of this study needs emphasizing particularly in the

More about the sport-specific background of internal and external load. How this can be predicted in sport and specific in soccer <(with or without the ball possesses.

Methods;

- pg.3 lin 101:

High-speed actions (HI) (all actions above 13 km/h) were analyzed in detai

My question: what about action form 0 to 13km/h)…

Results:

This is more a comment than anything - - can you think of a different way to present the data?

Conclusions:

The authors should mention also all the limitations of this study. The conclusion is written too generally.

Author Response

We sincerely appreciate the feedback provided by Reviewer 1. We have tried our best to make all the improvement suggested

Introduction:

- pg 2

The "why?" of this study needs emphasizing particularly. More about the sport-specific background of internal and external load. How this can be predicted in sport and specific in soccer <(with or without the ball possesses.

Thank you very much for the feedback. In addition to talking about the external and internal load in soccer, a new paragraph has been included in the introduction to help the reader to focus on the topic of the article.

Methods;

- pg.3 lin 101:

High-speed actions (HI) (all actions above 13 km/h) were analyzed in detai

My question: what about action form 0 to 13km/h)…

We appreciate this comment. It is a mistake we probably pressed 3 instead of 6 when writing this paragraph (3 is just below 6 in most keyboards). What we did was to analyse in more details those actions performed at high intensity. That means all actions performed above 16 km/h (Zone 5 + Zone 6). We focus on these actions because they are the most relevant under a performance point of view

Results:

This is more a comment than anything - - can you think of a different way to present the data?

The way we present data has been used in other similar studies (i.e. López Fernández et al. (2019) The Journal of Strenght & Conditioning Research. Doi: 10.1519/JSC.0000000000002090. However, we have made some minor changes in format to facilitate the reading.

Conclusions:

The authors should mention also all the limitations of this study. The conclusion is written too generally.

We appreciate this comment. The conclusion has been modified. We have also mentioned the main limitations of the study

Reviewer 2 Report

This study is about different play forms in sub-elite soccer. A team each of U14, U16 and U18 played 4 different settings. The performance was accessed using GPS and the heart rate.

The analyzed parameters only correspond partially to the one presented in the introduction l34-36.

The same measured as for elite soccer players was used. Are these the right parameters for the sub-elite players? What is the aim of the training of these group? Make the step to the elite league? Health? Enhance the personal performance? Please discuss how these measures correspond to the parameters of this study.

The result section and discussion section are not aligned with the aim of your work. In the aim you write about the physiological influence of the game forms and not on the differences due to age.

Specific comments:

Abstract:

L10.Please avoid abbreviations in the abstract.

L16.Please do not state the manufacturer of the measurement device in the abstract.

Result, I am missing for each age group the influence of the game type. This is required to understand the conclusion.

L20-21 this sentence is not clear.

Introduction:

Method:

L56 please round the numbers (where ever appropriate) to 163 pm10. The other digits have no meaning.

Was for each group the ssg randomized? Or could the result be due to different motivation of the age groups?

Table 2. it is not clear to me if U14, total distance 3v3 is different to 6v6. If yes why is no “a” behind the 432.

Please round the numbers in the entire table for total distance and mean m/min. (all tables)

What is mean m/min? the unit is probably also m/min?

Discussion

L213-215. This seems to be a new topic. If the endurance training is the aim. The corresponding zones should be named and addressed.

Author Response

We sincerely appreciate the feedback provided by Reviewer 2. We have tried our best to make all the improvement suggested

The analyzed parameters only correspond partially to the one presented in the introduction l34-36.

Thank you very much for the feedback. In addition to talking about the external and internal load in soccer, a new paragraph has been included in the introduction to help the reader to focus on the topic of the article.

The same measured as for elite soccer players was used. Are these the right parameters for the sub-elite players? What is the aim of the training of these group? Make the step to the elite league? Health? Enhance the personal performance? Please discuss how these measures correspond to the parameters of this study.

We sincerely appreciate this comment. Sub-elite academies at least in Spain where the study was conducted are focused on performance although at a lower level. Therefore, they focus on improving fitness condition, technical and tactical skills of their players, but also to classify as best as possible in the league. These objectives are relatively similar to the elite academies. The point of selecting a sub-elite academy is that the fitness and technical-tactical level of players is by definition lower so designing the SSGs according to the findings from top players might be inadequate. Same would happen if we use the findings from male soccer training to design female soccer training. This point is addressed in the introduction (line 56 to 58).

On the other hand, we understand your concern about the measures used in this study. However, they are similar to previous studies with other players level in order to provide comparisons. In this regard, it is also important to consider that there is not enough information to justify using different measures. Therefore, to reduce the risk of bias and facilitate comparisons we decide to use these measures. However, we do acknowledge the importance of your comment, so we have added this fact into the work’s limitation.

The result section and discussion section are not aligned with the aim of your work. In the aim you write about the physiological influence of the game forms and not on the differences due to age.

Thank you very much for your feedback. The aim of the research has been modified accordingly

Specific comments:

Abstract:

L10.Please avoid abbreviations in the abstract.

Thank you very much for the comment. It has been modified accordingly

L16.Please do not state the manufacturer of the measurement device in the abstract.

Thank you very much for the comment. It has been modified accordingly

Result, I am missing for each age group the influence of the game type. This is required to understand the conclusion.

Thank you very much for the comment. It has been modified accordingly

L20-21 this sentence is not clear.

The sentence has been rewritten.

Method:

L56 please round the numbers (where ever appropriate) to 163 pm10. The other digits have no meaning.

Thank you very much for the suggestion. We have rounded the numbers accordingly

Was for each group the ssg randomized? Or could the result be due to different motivation of the age groups?

No, all aged groups followed the same order. However, to reduce the influence of both fatigue and motivation as well as to increase data reliability first the players followed the order: 3v3, 4v4, 5v5 and 6v6. The second-day test they follow the inverse order 6v6, 5v5, 4v4 and 3v3. This was because we only had access to two days test + Yo-Yo Intermittent Endurance Test Level 2 + the familiarization session (in total 4 days). However, we acknowledge your comments and it this fact has been included as a limitation of the study

Table 2. it is not clear to me if U14, total distance 3v3 is different to 6v6. If yes why is no “a” behind the 432.

To facilitate the reading, we decided to identify significant differences only in the value with a higher mean. In this case, Table 2 is showing significant differences between 3v3 and 6v6 in U14. But it is pointed out in the 3v3 column with a “d”. This is because the mean for 3v3 is higher (464.15) than the mean for 6v6 (432.11).

Please round the numbers in the entire table for total distance and mean m/min. (all tables)

Thank you very much for the suggestion. We have rounded the numbers accordingly

What is mean m/min? the unit is probably also m/min?

Thank you very much for the comment. It is the mean of m covered per minute. As you say the units are m/min. It has been modified accordingly.

Discussion

L213-215. This seems to be a new topic. If the endurance training is the aim. The corresponding zones should be named and addressed.

This paragraph has been rewritten. We appreciate your feedback.

Reviewer 3 Report

The Article “PHYSICAL AND PHYSIOLOGICAL RESPONSES OF U-14, U-16, AND U-18 SOCCER PLAYERS ON DIFFERENT SMALL-SIDED GAMES“ is interesting and relevant recently. The article is well and quite clearly written but it requires a number of changes before it will be published:

  1. In the Summary part there is presented abbreviations without explanations (e.g. SSGs), it would be more informative to reminded in notes what does abbreviations mean.
  2. It is not clear how many measures were done for each subject? And why some measures were deleted.
  3. I would like to suggest to some tables (e.g. 1, 2, 3 table).
  4. There was not explained when the data collection was collected, in what season, month or etc.
  5. Data in results part presented in a tables or figures but I have to note that their description is a bit insufficient – to much repetitive data and too much labels in the text.
  6. I would like to try to find some interrelations between physical and physiological features (or as it cold in the article between internal and internal loads).
  7. Conclusions in my opinion, written not clear, particularly in the Abstract.

Before publication, in my opinion, article has to be improved.

Author Response

We sincerely appreciate the feedback provided by Reviewer 3. We have tried our best to make all the improvement suggested

In the Summary part there is presented abbreviations without explanations (e.g. SSGs), it would be more informative to reminded in notes what does abbreviations mean.

Thank you very much for your suggestion. Changes have been applied accordingly

 It is not clear how many measures were done for each subject? And why some measures were deleted.

Thank you very much for the comment. We have provided further explanation in the manuscript. In total each player was recorded two times per SSG format. Regarding the measures deleted, the GPS device failed and only provided partial measures.

I would like to suggest to some tables (e.g. 1, 2, 3 table).

We think there are some information missing. If required, we are happy to address the corresponding changes in a future revision

There was not explained when the data collection was collected, in what season, month or etc.

Thank you very much for the comment. We have provided the requested information in the manuscript

Data in results part presented in a tables or figures but I have to note that their description is a bit insufficient – to much repetitive data and too much labels in the text.

Thank you very much for the comment. We have tried our best in improving this section. All descriptions have been reviewed and improved accordingly.

I would like to try to find some interrelations between physical and physiological features (or as it cold in the article between internal and internal loads).

We sincerely appreciate this feedback. However, it is very difficult to make interrelation between physical and physiological variables if a correlation or regression is not performed. In this regard, we can make some assumptions but somehow, they could make bias in the reader.

Conclusions in my opinion, written not clear, particularly in the Abstract.

We appreciate this opinion. The conclusion has been rewritten in both the Abstract and the Main Document.

Round 2

Reviewer 2 Report

thank you for the revision. all my comments have been addressed properly.

Author Response

The authors thank the Reviewer's two for the provided feedback. We have addressed the final improvements suggested by the editors

1) All references have been included in the text. I did not realise that some of them were still on comments. I am very sorry for this mistake.
2). The English grammar has been reviewed by an English college we hope it is now good enough.
3) Figure 1 has been amended. We are not sure about when the symbols vanished, but we have re-attached the Figure 1 with the symbols included.
4) The tables foot and Figures foot has been improved according to the suggestion

We sincerely appreciate your feedback